# Reproductive Potential of Stone Moroko (*Pseudorasbora parva*, Temminck et Schlegel, 1846) (Teleostei: Cypriniformes: Gobionidae) Inhabiting Central Europe

**DOI:** 10.3390/ani11092627

**Published:** 2021-09-07

**Authors:** Lucyna Kirczuk, Katarzyna Dziewulska, Przemysław Czerniejewski, Adam Brysiewicz, Izabella Rząd

**Affiliations:** 1Department of Hydrobiology, Institute of Biology, University of Szczecin, Felczaka 3c Street, 71-412 Szczecin, Poland; lucyna.kirczuk@usz.edu.pl; 2Molecular Biology and Biotechnology Centre, University of Szczecin, Wąska 13 Street, 71-415 Szczecin, Poland; izabella.rzad@usz.edu.pl; 3Department of Commodity, Quality Assessment, Process Engineering and Human Nutrition Westpomeranian University of Technology in Szczecin, Królewicza 4 Street, 71-550 Szczecin, Poland; przemyslaw.czerniejewski@zut.edu.pl; 4Institute of Technology and Life Sciences, National Research Institute, Falenty, 3 Hrabska Avenue, 05-090 Raszyn, Poland; a.brysiewicz@itp.edu.pl; 5Institute of Marine and Environmental Sciences, University of Szczecin, Wąska 13 Street, 71-415 Szczecin, Poland

**Keywords:** fecundity, gonadal histology, stone moroko, sexual cycle, invasive species

## Abstract

**Simple Summary:**

Stone moroko (topmouth gudgeon, *Pseudorasbora parva*) is a small fish that originates from the East Asia. The fish is an invasive species that extends its global distribution. The study aimed to assess the reproductive potential of the stone moroko in a new habitat in central Europe based on an analysis of the sexual cycle. A rapid sexual maturation and high-fecundity spawning in portions prolonged the four-month reproductive period, which enables a rapid increase in population size and density on a newly inhabited area. Furthermore, the smaller investment in gonad development in males allows them to retain vitality for nest protection and a longer reproductive season compared to the autochthonous cyprinid fish. The present study suggests that stone moroko population in Central Europe may exhibit a higher reproductive potential than native populations.

**Abstract:**

Similar to other invasive species, stone moroko is extending its global distribution. The present study aimed to assess the reproductive potential of stone moroko fish in a new habitat in Poland based on analysing the sexual cycle and fecundity. Fish morphometric data, age, and gonadal structures were analysed. Fish age ranged from 0+ to 5+ years. Most females and males (93% and 60%, respectively) had reached sexual maturity in the first year of their life, with the smallest length of 25 mm and 28 mm, respectively. The mean, standard length of the body was 50 mm. The spawning season was spread over four months from late-April to mid-August. Females laid eggs in portions, and the absolute and relative fecundity was 1372 and 1691, respectively. Stone moroko males were ready to spawn for a longer time period than females. The present study shows greater reproductive potential of stone moroko population in the central Europe than the native population, suggesting its successful colonisation in the new habitat.

## 1. Introduction

Stone moroko (topmouth gudgeon, *Pseudorasbora parva*) is a small fish from the Cypriniformes, originally from East Asia. It is found in different habitats, most commonly small channels overgrown with vegetation, pools, and small lakes [1,2]. Due to its rapid dispersal, negative impact on native hydrobionts, and transfer of infectious diseases (*Sphaerothecum destruens* (*Protista*: *Mezomycetozoea*)), the fish is considered an international pest species in Europe [3,4]. The stone moroko was first found in Europe in Romania and Albania in 1961 [5,6]. Over the following years, it spread throughout Europe, as well as North Africa (Algeria, Morocco), Middle East (Turkey, Iran), Central Asia (Kazakhstan, Uzbekistan), and even Oceania (Fiji) [4,7,8,9]. The major vector for the dispersal of this species was its accidental introduction with stocking material of other fish species to ponds, rivers and lakes. It was first recorded in Poland in 1990 [10], and it is currently widely distributed in the country [11,12]. The stone moroko has been classified as a potential pest and is considered one of the most invasive foreign fish species [13,14]. Its distribution is facilitated by its wide feeding spectrum, substantial environmental tolerance, and small size [2,15,16,17]. In carp pools, particularly during the mass occurrence, it competes with the breeding fish species [18,19]. In the open waters and pond breeding facilities of southern Europe, the stone moroko likely contributed to reduced numbers and even to the disappearance of certain autochthonous cyprinid fish species, such as common rudd (*Scardinius erythrophthalmus*), crucian carp (*Carassius carassius*), Amur bitterling (*Rhodeus sericeus*), gudgeon (*Gobio gobio*), and sunbleak (*Leucaspnis delineatus*), by feeding on their eggs and juvenile stages [20,21]. In greater population densities, it may pose a severe threat to rare and endangered fish species [22], reduced biodiversity [23], and transmitted infectious diseases [16]. The effects are similar to those of other invasive fish [24].

The pronounced expansion of this species is also caused by reproductive traits of this species related to laying eggs in portions, high fecundity of females, or egg guarding by males [15,16,17,25]. Research on the reproductive traits of this species has been conducted rarely in its natural distribution area, as well as on recently invaded continents [26,27,28]. This kind of study has been typically carried out on females, and the assessment of developmental stages was performed based on the external appearance and weight of gonads without detailed histological analysis [27]. Samples from selected months were analysed [29], females maintained in laboratory conditions were studied [30], and only selected microscopic characters of oocytes were described [31]. Thus far, only the Japanese population was subjected to a more thorough analysis of the histological structure of gonads and sexual cycle [26], whereas the populations introduced to new areas have been only shown to have a high reproductive potential [29,32]. The present study aimed to (I) analyse the reproductive cycle of both stone moroko sexes based on microscopic observations; (II) assess the reproductive potential of a population; and (III) assess its potential influence on the ‘species’ expansion in a new environment based on a small watercourse in Central Europe.

## 2. Materials and Methods

### 2.1. Location and Sampling

The fish were collected from the Wardynka River (north-west Poland, Figure 1) of the Oder basin by electrofishing, once a month from September 2019 until August 2020, in the reproductive season every two weeks (from May to August), each time by about 30 individuals. A total of 602 fish were collected (Table 1). The fish were euthanised with an overdose of MS-222 (300 mg L^−1^, Sigma-Aldrich). The fish’s body length (total length TL and standard length SL) was measured to the accuracy of 1 mm. The fish were weighed whole and eviscerated on an electronic scale to the accuracy of 0.1 g. Gonads were prepared and weighed with 1-mg accuracy. In the period from late-April to July, the right and left gonad of females was weighed for fecundity analysis. Scales above the lateral line on the left side of the body were collected. Fish age was determined based on scale analysis [3,33]. The Fulton condition index (C_F_ = body mass (g) × length SL^−3^ (cm) × 100%), gonadosomatic index (GSI = gonad weight (g) × fish body weight ^−1^ (g) × 100%), and GSI of eviscerated body weight were calculated.

### 2.2. Physicochemical Parameters of Water

For sampling, measurements were taken directly in the field. Temperature, conductivity (EC), and water oxygenation were measured with multiparameter sensor HQD30 produced by Hach. At the same time, water samples were taken from the rivers following the current standards (PN-EN ISO 5667-6:2016-12, PN-EN ISO 5667-3:2018-08) to determine the concentration of N-NO_3_, N-NH_4_, and P-PO_4_. The concentration of nitrogen forms and phosphates was determined colourimetrically with an automatic flow analyser produced by Skalar.

### 2.3. Histological Slide Preparation and Fecundity

Female and male gonads were fixed in 8% formalin or Bouin fluid, respectively, and histologic preparations were developed with the standard paraffin technique. Five-µm-thick sections were regressively stained with Heidenhain’s iron hematoxylin or Mayer’s hematoxylin and eosine. One microscopic slide contained 5–50 snips of the gonad. The histological slides were analysed, and oocytes were measured under a Nikon Eclipse 80i light microscope. The photographic documentation was made using the NIS Elements 3.20 program and a Nikon DS-5Mc-U2 digital camera with of 5-M pixel resolution. The sexual cycle of female gonad was described using the modified 6-grade scale applied in Domagała et al. [34]. Oocytes developments were divided into five size classes: group 1-oocytes with completed vitellogenesis (fat joined with yolk), group 2-oocytes with finalising vitellogenesis (oocytes with yolk grains), group 3-oocytes at initial vitellogenesis (oocytes with lipid droplets), group 4-oocytes in advanced previtellogenesis, and group 5-oocytes in early previtellogenesis (supply for the next season). The diameter of 30–50 oocytes from each class was measured under Nikon Eclipse 80i. The oocyte diameter was calculated from measurements of the longest and shortest diameters of the mid-cross-section of the oocyte at 0.01-μm accuracy.

The sexual cycle of males was described using a modified 6-stage scale published in Domagała et al. [35]. For the stone moroko cycle, the overlapping of the gonadal cycles as early as in stage V was designated as substages V-I and V-II.

The number of oocytes in the left gonad were counted under the Discovery V12 Zeiss stereoscopic microscope. Subsequently, the number of oocytes from each size group was counted per each of two gonads using the gravimetric method [36]. Absolute fecundity was calculated based on the number of oocytes from groups 1–4. Absolute fecundity did not include the smallest oocytes from group 5, because they do not mature in the current season [37]. Relative fecundity was calculated as the number of oocytes per 1 g of eviscerated body weight.

### 2.4. Statistical Analysis

Prior to the analysis, the normal distribution of variables was tested using the Shapiro–Wilk test and homogeneity of variance using the Levene test [38]. The one-way ANOVA test was used to compare the following characteristics of the fish: body length and mass, condition coefficient (C_F_), gonad mass, gonadosomatic index (GSI), and oocyte size. The significance of differences in the number of females and males was tested with the chi-square test. The Pearson correlation was calculated between the number of oocytes groups 1–4 (absolute fecundity) and the females’ weight, and length, and age of females. The same correlation was calculated for the number of oocytes per 1 g of eviscerated body weight (relative fecundity). All analyses were performed at the significance level of 0.05 using the Statistica v.13.1 software (Dell Inc., Tulsa, OK, USA).

## 3. Results

### 3.1. Physicochemical Parameters of Water

The lowest water temperature was 4.9 °C (February), and the highest was 24.1 °C (July). The highest oxygen content was observed in the winter months (9.41 mg L^−1^ in January) and lowest in July at 5.19 mg L^−1^ (Figure 2). Mean water flow rates were 0.54 ± 0.17 (0.41–1.07) m s^−1^; N-NO_3_ content 1.81 ± 0.75 mg L^−1^ (0.92–3.11), N-NH_4_ 0.33 ± 0.17 mg L^−1^ (0.04–0.57), and P-PO_4_ 0.95 ± 1.06 mg L^−1^ (0.10–3.35).

### 3.2. Length, Weight, C_F_, Gonad Weight, GSI

#### 3.2.1. Female

The ratio between the number of females to males collected within the study was (0.62:1). Female age ranged from 0+ to 5+. SL and weight of *Pseudorasbora parva* females was 46.2 ± 12.4 mm and 1.85 ± 1.66 g, respectively (Table 2).

The highest and the lowest GSI of females was recorded in May and January with a mean of 8.55 (0.53–25.68) and 1.79 (0.21–2.91), respectively (Figure 3). The GSI of females were 5.61 (3.18–8.62), 12.33 (8.23–25.68), and 2.79 (0.75–4.60) at pre-spawning, spawning, and post-spawning times, respectively. The considerable breadth of spawning female GSI stems from the fact that some females were after laying a portion of oocytes and prepared (at a different stages) to lay further portions. GSI eviscerated body weight in females were 6.51 (4.26–9.67), 13.94 (8.98–31.55), and 3.29 (1.03–8.30) at pre-spawning, spawning, and post-spawning times, respectively.

#### 3.2.2. Male

Males caught were aged 0+ to 5+. Individuals belonging to this sex were longer than females; the mean, standard length (SL) of males was 52.9 ± 14.1 mm and weight of 3.30 ± 2.72 g (Table 2).

Gonadosomatic index changed throughout the year with gonad maturation and sexual cycle. The highest GSI in males of a mean value of 2.4 was recorded at the beginning of the breeding season in the second half of May. In this period, in individuals finalising spermatogenesis (stages III_L_ to IV), the GSI ranged from 0.99 to 6.93. In June and July, different amount of sperm was ejaculated/produced, and various stages of maturity were detected (stages IV, V, VI and II, I), causing a decrease in mean GSI to the level of 1.0–1.5. In the subsequent month, gonads in stage II predominated, and the mean value of the index remained at a similar level 0.53–0.88 (from August to March). A few individuals, after spawning, beginning a new cycle from stage I of mean GSI of 0.42. In April, GSI increased to 1.2 with relation to the appearance of gonads at the III_L_ stage (Figure 3). 

### 3.3. Sexual Cycle

#### 3.3.1. Female

From September to February, all gonads of stone moroko females were in stage 3 (Figure 4A), and some gonads were observed to contain degenerating oocytes at different stages of vitellogenesis (Figure 4B,C). In March, 20% of females entered stage 4. In April, the majority of females had gonads characteristic of an individual preparing for spawning, and one female laid the first portion of eggs. In mid-May, 50% of females spawned, and at the end of the month, all females laid eggs and were at different stages of preparation with a greater or lower number of oocytes for laying subsequent portions. In mid-June, all females were laying subsequent egg portions. At the end of the month, most females were still laying eggs, but some of them were after spawning, with gonads without further portions of oocytes or with developing stage 3 for the next season (Figure 4D). In July, almost 40% of females were still laying eggs, whereas in August, only a remaining few had post-spawning gonads with degenerating oocytes, which have not been removed after spawning (Figure 4E). In this month, singular gonads were observed at stage 2 with degenerating oocytes at the stage of previtellogenesis (Figure 4F) and stage 3 (Figure 5). In the period from April to July, 7% of one-year females did not attempt spawning, i.e., those remaining at stage 3. The smallest female trying spawning in the studied sample was 25 mm (SL).

##### Oocyte Appearance and Size

Stone moroko spawning comprises multiple portions, and females lay 2–3 portions of eggs per season. The first portion of eggs was laid by stone moroko females in the second half of April, which is indicated by the presence of post-spawning gonads. Dimensions of the largest oocytes point to laying subsequent two portions of eggs in the second half of May and in the second half of June. In terms of size, five oocyte groups were distinguished for stone moroko females. The statistical test revealed that differences between oocytes were significant in all age groups. However, oocyte sizes in all groups of fish collected from April through the end of May did not differ significantly (*p* > 0.05) (Table 3).

In females collected until mid-June, a lower oocyte diameter was observed in individual groups, which may be linked with the deposition of subsequent egg portions (Figure 6).

##### Fecundity

The mean absolute fecundity in all age groups of stone moroko is 1372 eggs (range 225–6620), while relative fecundity is 1691 eggs per 1 g (range 268–4543) (Table 4).

With age (Figure 7A) and body length (SL) (Figure 7B), females had an increasing number of oocytes in gonads (absolute fecundity) (Pearson coefficient 0.81 and 0.75, respectively). However, a negative correlation between relative fecundity and the weight and length and age of females was obtained.

The mean number of smallest oocytes (group 5) amounted to 1466 (222–8212). A significant correlation coefficient between the number of the smallest oocytes and fish age (0.55) and an insignificant correlation between the number of these oocytes and SL (0.07) were observed. Among the analysed females, two individuals had a particularly high number of oocytes in early previtellogenesis. A female aged 4+ (SL 6.4 cm, collected at the end of June) had 12548 oocytes of group 5, and 5220 from groups 1–4, while another female at the same age (SL 62 mm, collected in July) had 23653 and 3649 oocytes, respectively. The largest stone moroko female aged 5+ (SL 78 mm collected in April) had 3027 of group 5 prior to spawning. The females aged 3+ had the highest number of oocytes from classes 1–4 (6620).

#### 3.3.2. Male

In autumn and early winter, the stone moroko male gonads were at stage II of maturity (Figure 5B). Seminiferous tubules of the testis were filled with spermatogonia A and cyst containing spermatogonia B. Most males reached late substage II (85%, II_L_) in the period with cyst containing from 10 to 20 spermatogonia B (Figure 8A). The remaining males (15%) were in early substage II_E_ with less than ten spermatogonia B in the cross-section of the cyst. In January, the number of type B spermatogonia in the cyst increased up to 40 cells, and some individuals reached early-stage III_E_ (20%) (Figure 8B). In the latter stage, the meiotic division has begun, and primary spermatocytes were formed in a cyst. In February and March, the histological image of the gonads was similar as in the previous month (Figure 4B). In April, individuals at very different stages of maturity from stage II_L_ to stage IV were observed. In the individuals at stage III_L_, the first spermatozoa appeared (Figure 8C). In late April, ripe gonad at stage IV containing spermatozoa in the lumen of the seminiferous tubules and the efferent ducts were found in 13% of males. Gonads’ cysts with spermatogonia and spermatocytes were still numerous at the tubule wall (Figure 8D). In May, individuals from III_E_ to IV stages were observed. In mid-May, spawning males constitute 30%, while in late May, they predominate (92%). In June, mature individuals at stage IV or V were less numerous (75%) (Figure 5B). In gonads at stage V, spermatogenesis in the seminiferous tubules was completed, and the lumen of the tubules was filled with spermatozoa. At the tubule wall, spermatogonia A or, in most males, spermatogonia B were multiplied (stage V-II) (Figure 8E). In that month, few individuals with spent gonad at stage VI-II (Figure 8F) and stage II were also recorded. In July, two-thirds of males were still ready for reproduction (stages IV and V-II), while the remaining males were at stage VI-II or II. In August, a few males were leaking with milt (15%, stages IV and V-II) or had spent gonad (20%, stage VI-II). The dominant group were males at stage II (40% at substage II_E_ and 15% at II_L_). A few males with gonads going down to stage I was recorded (10%). In those months in gonads, a large number of degenerating cells were visible. The features were numerous until the end of October. In September, some males still had ripe gonads (7%, stage IV), while most males were at stage II_L_ (70%). The remaining males were at stage I or II_E_. In the following months, from October to December, most individuals were in late substage II_L_ (85%) while a minority was in early substage II_E_ (15%). In individual males caught in Wardynka River, a small number of spermatozoa from the completed reproductive season was found in seminiferous tubules until February. In the studied stone moroko population, as many as 4% of males had few oocytes in the testicular tissue at the previtellogenic stage. Detailed percentages of males with gonads at each maturity stage during the calendar year are shown in Figure 5B.

The smallest male attempting spawning in the studied sample was 28 mm (SL).

## 4. Discussion

Stone moroko individuals collected from the Wardynka River belong to non-native populations inhabiting the Oder River basin. Among these populations, the population from the Wardynka River is probably characterised by the highest density (1.77 individuals/m^2^) [39]. The first record of the ‘species occurrence’ in the Wardynka River was published in 2019 by Czerniejewski et al. [39] on a locality in its upper course. The fish obtained for the present study likely originate from the same population, which has invaded a new area through expansion downstream of the river over the last several years.

Stone moroko is a short-lived species [2,40]; therefore, the maximum fish age in the tested population was 5+. On other sites, both from the native distribution area as well as newly populated habitats, stone moroko typically reached the age of 3+ [27], 4+ [40,41,42], as well as 5+ [17,27]. In the analysed population, the highest number characterised the youngest specimens, indicating a developing population. Water temperature, access to food, predators, and stress affect the population age structure [27,43]. Good physicochemical conditions, accessibility of food, and a lack of predators in the Wardynka River contributed to the colonisation success of this population and its prolonged viability [33].

The sex ratio in stone moroko populations in different areas of distribution varies. In the studied *Pseudorasbora parva* population from the Wardynka River, males were predominant (females: males 0.62:1). In other areas of Poland and Chinese lakes, females were less numerous than males at 0.18:1 [27,44]. The sex ratio was rarely equal to 0.99:1 [27] or with female predominance [34]. In other cyprinids, fish females were mostly predominant [35,45,46,47,48,49,50], although equal proportions of the sexes or domination of males were recorded [51,52,53,54,55,56,57].

The mean SL of the fish obtained for the present study was 50.0 mm (range 25–87 mm). Males from Wardynka were larger than females, as shown in other studies [44,58,59]. However, stone moroko from Slovakia were smaller (mean of 32.6 mm, range 18.1–67.5 mm) [40]. Similarly, smaller individuals were collected in Turkey (fork length females 14.7 mm, males 50.9 mm) [59]. However, in certain Polish populations, the mean length was greater (66.5 ± 4.5 mm), which is probably related to the faster growth of these fish in culture ponds [44].

The smallest maturing individual of the tested population was a one-year female measuring 25 mm SL. The smallest maturing male was 28 mm. Only in Slovakia, smaller sizes at maturity were observed. The smallest mature female measured 23.4 mm [28,40]. Indeed, all specimens over 26-mm SL were found to have fully developed gonads [32]. In the tested populations, females mature very early, after the first winter before the first annulus was laid down. In the tested habitat in Poland, 93% females and 60% males mature in the first season. Similarly, in the Slovakian population, the lower mean age at maturity of 0.9 years was in females than 1.1 years in males, and 94% of specimens from the age group I were already mature [40]. In the Netherlands, adults’ individuals were >24 mm fork length, L_F_ [8]. In China, all individuals matured at age 1+ at minimal size 36.1 mm [27]. In Europe, cyprinids females reach sexual maturity at the age of 2 or 3 and older [35,48,60]. In more southern and warmer climates of its distribution, sexual maturity is reached at a younger age in the first year of life [53]. Typically, males reach maturity a year earlier than females [48,60,61,62].

The estimated total fecundity of stone moroko included counting four groups of largest oocytes (with the exclusion of the smallest oocytes, group 5) which correspond in terms of size to three groups of oocytes from the study of Záhorská and Kováč [29]. The mean total fecundity of stone moroko from Wardynka was 1372 oocytes and the relative fecundity was 1691; this was greater than in this species from Slovakia, for which 1106 and 1419 oocytes were recorded, respectively [29]. A total of 1576 (639–2816) were found in Danube Delta, Romania [20], and 388 (93–1400) in Amur River Basin [63]. Maturation of 2–3 portions of eggs was observed in the present study. Other researchers have shown 3–4 portions per season [2]. In the study on stone moroko, fecundity under laboratory conditions, from 167 to 6285 eggs, was observed, laid in 1 to 14 mating sequences [30]. Our study shows a positive correlation of absolute fecundity with length and age of the fish and a negative correlation relative fecundity z with the weight and length and age of females, similar to nase [64]. The aforementioned studies did not refer fecundity to female moroko age. Stone moroko has a high relative fecundity (number of oocytes per 1 g eviscerated body weight), as compared with other Cyprinidae, such as white bream 70–140 [65], 110 blue bream [66], and 186 roaches [67]. Dimensions of mature oocytes of stone moroko from the Wardynka River were in the range between 0.07–1.73 mm and were similar to the data of Zhu et al. (0.08–1.27 mm) [28]. The diameter of the largest oocytes in the examined fish was 1026 µm (795–1725 µm). In the study of Záhorská and Kováč [29], this value was 1130 µm (350–1560 µm). Disturbance of the environment inhabited by stone moroko and a recent invasion (<5 generations) resulted in an increased number of oocytes and their reduced size, with younger individuals maturing [32].

The spawning date depends on local conditions, photoperiod, and temperature [28,39,68]. The course of the sexual cycle in the tested species was determined so far on the basis of the outer appearance and mass of the gonad or gonadosomatic index [27,29,30]. Histological analysis of the sexual cycle in stone moroko was only performed in a native Japanese population from the Tamam River, Tokyo [26], which enables detailed observation of the changes occurring in gonads and comparing the gametogenesis dynamics with invasive populations present in new habitats. Based on the histological tests of the investigated stone moroko population from north-west Poland, it was established that the reproductive season of the species is long, stretching over the period of 4 months. Based on the gonad structure, it was determined that the reproductive season commenced at the end of April and lasted till mid-August at a temperature from 14 °C to 24 °C (April–July) with a peak between the second half of May, going into June. In its native Amur River, stone moroko spawned at a temp. of 15–19 °C [63]. The reproductive season in its native range in China and Japan lasted three months from March to June (April–July) [26,27]. It also lasted three months in the invaded areas of southern Europe, from April-June (Romania) [20]. It was longer in Czechia from March to June [69], while in Ukraine, it took place from May through September [19]. In a colder Lake Fuxian in China (max temp 22.4 °C), there was a delay of spawning time and extending spawning duration to 5 months from April to August [27]. In Poland’s studied invasive population of stone moroko, the spawning period lasted four months and was considerably longer than native cyprinids. Depending on the species, the spawning time in these fishes’ ranged between 2 weeks and 2–3 months. However, the reproductive season of native cyprinids in Poland typically fell on a similar period as for stone moroko: May–June [35], i.e., it commences less frequently in April [70].

Males of studied stone moroko were ready to spawn at the end of April, at the same moment as the females. In other cyprinid species, males became ready for reproduction earlier than females [35,49,71]; only a few species are ripe at the same moment as females [50]. Similarly, to other cyprinids, during the peak of reproduction, gonads of male stone moroko are at stage IV, when the successive formation of new sperm cells occurs as a result of maturation of propagating new cells of the spermatogenesis process. Similarly, to other cyprinid species, stone moroko males maintained the ability to spawn for a longer period than females [35,49,71]. In the case of stone moroko, about 10% of males retained reproduction capability for a month longer than females. At the end of spawning, two groups of males were distinguished: mature and post-spawning. In the Japanese population, two groups of males differed in the period of recrudescence of gonads were observed [26]. In the population, spermatozoa from the completed cycle remained in the gonads until September [26]; however, in the analysed Polish population, small quantities of not removed spermatozoa were visible until the end of February. The period of meiosis commencement constituted a marked difference in the sexual cycle between these two areas.

In Japan’s native distribution area, primary spermatocytes were observed in autumn, but active spermatogenesis never occurred until next spring [26]. In the Polish population, spermatocytes were first observed as late as January, and propagation of B spermatogonia was only observed in autumn and winter (stage II_L_). During autumn and early winter, gonads in other cyprinids are also at the II_L_ stage; only in blue bream during winter do gonads attain stage IV containing even spermatozoa [35,49,50,71,72]. In the latter species, stage II is short, and spawning commences at the earliest time among cyprinids, i.e., from the beginning of April. In Poland, stone moroko was observed to have early propagation of B spermatogonia for the next season already at the finalisation stage of the current spermatogenesis (substage V-II), moving then to stage VI-II and II along with sperm cell resorption. Few males (around 10%) had gonads return to stage I after spawning.

In contrast, typically, the propagating B spermatogonia commenced the preparation for the next cycle already at stage V. Perhaps the high temperature and short photoperiod are responsible for the cessation of the spawning period in *Pseudorasbora parva*, similar to other cyprinids [26]. In the period of completed spawning, a high number of degenerating cells was observed for both males and females. Intensification of the process in males was observed until the second half of September and October when cells of probably active spermatogenesis degenerated. In female gonads, degenerating previtellogenic and vitellogenic oocytes were observed from June. The degeneration of vitellogenic and post-spawning oocytes is typical of *Cyprinidae*, which was previously recorded for white bream and blue bream [35,49]. Asahina [26] only observed degeneration of the oocytes in stone moroko that were not expelled during spawning in July.

The GSI fluctuation on an annual cycle corresponds to the changes occurring in gonads. The highest GSI of the analysed stone moroko females was maintained from April through to June, reaching a mean value of 12.3 (maximum individual GSI was 25.7). Females from the Japanese Tama River who had similar GSI values were noted from April through June, with a peak of 16.0 observed in June [26]. In other cyprinids, such as bream, white bream, and blue bream, the GSI of spawning females attained similar as in stone moroko. The highest values were recorded in April and May, in the range from 12.0–18.0 [35,49,70]; only in rudd was it slightly lower at 10.5 [50].

The highest GSI in stone moroko males was also recorded at the beginning of the reproductive season, with a mean of 2.4. The maximum GSI in Japanese stone moroko was also 2.0 [26]. In other cyprinids, the GSI in males attained higher values, up to the mean of 6–7 in rudd and white bream, i.e., only half lower than females [35,50]. The GSI of the analysed stone moroko males was five times lower than the females (2.4 and 12.3%, respectively). Lower investment in gonads in stone moroko males might affect the prolonged reproductive capacity and agility in the care of the nest, contributing to the species’ reproductive success.

In the studied stone moroko population, as many as 4% of males had few oocytes in the testicular tissue at the previtellogenic stage. Similar inclusion of oocytes in gonochoristic species has been frequently identified in numerous fish species [73,74,75,76]. Perhaps such a phenomenon is common because the genetic grounds for sexuality in fish is very labile and diverse. Thus, the environmental conditions, such as incubation temperature and early development, as well as water pollution with heavy metals, organophosphates, organochlorines, and compounds found in pesticides, herbicides, and steroid derivatives of other industrial waste, may depend on the concentration, affecting the activity of enzymes, gene expression, level of steroid hormones, sex determination, sex reversion, or other disturbances related to reproduction [73,77,78].

## 5. Conclusions

Our studied population found in a newly invaded habitat exhibited a higher reproductive potential than native populations of stone moroko. A rapid sexual maturity characterised the fish that were already in the first year of life (females in particular), and caused high fecundity, spawning in portions, and a prolonged, four-month reproductive period, enabling a rapid increase in population size and density of a newly inhabited area. Furthermore, the smaller investment in gonad development in males allowed them to retain vitality for nest protection and a longer reproductive season compared to other native species of cyprinids. The reproductive potential combined with minor habitat requirements and the capability to adapt to different environmental conditions contributes to the colonisation success of this species in a new area.

## Figures and Tables

**Figure 1 animals-11-02627-f001:**
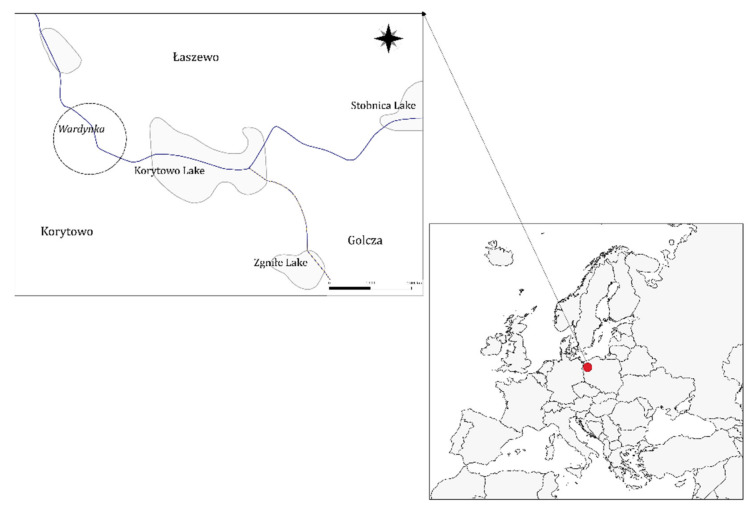
Research area in the Wardynka river, 53°08′47.0″ N 15°34′00.8″ E.

**Figure 2 animals-11-02627-f002:**
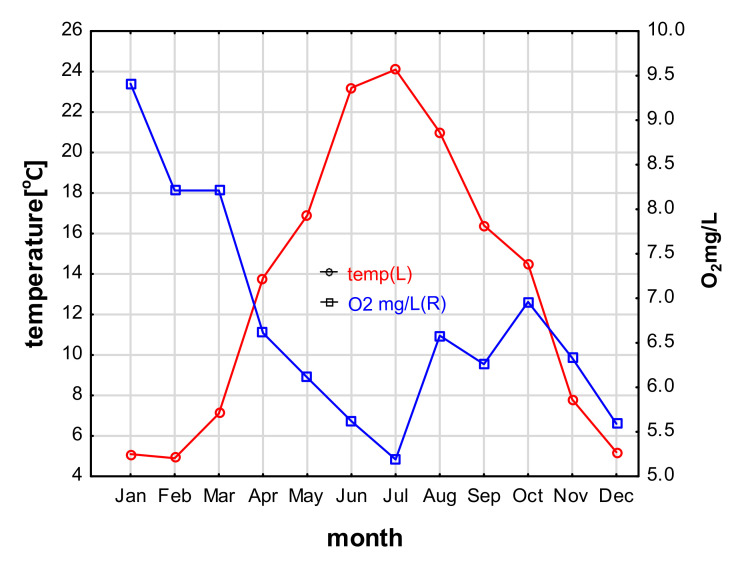
Temperature and oxygen content in the Wardynka river (Poland) in the years 2019–2020.

**Figure 3 animals-11-02627-f003:**
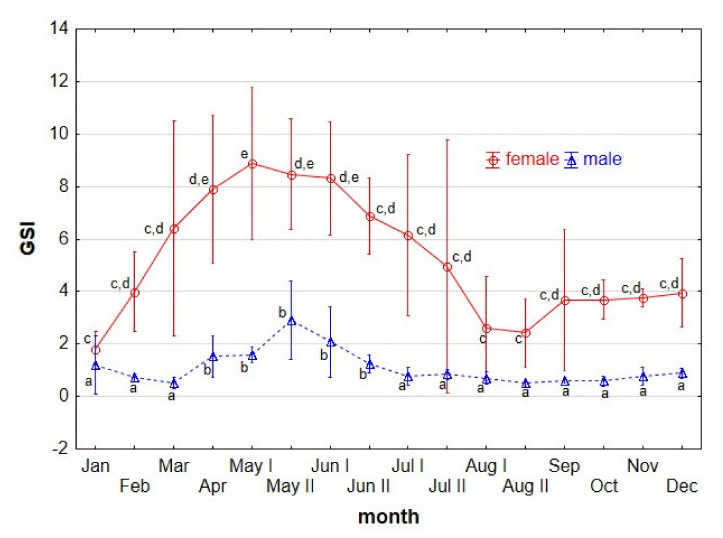
Monthly changes of gonadosomatic index (GSI) of stone moroko. (*Pseudorasbora parva*) female and male inhabiting the Wardynka river, Poland. Values with different letters show significance differences (*p* < 0.05; ANOVA test).

**Figure 4 animals-11-02627-f004:**
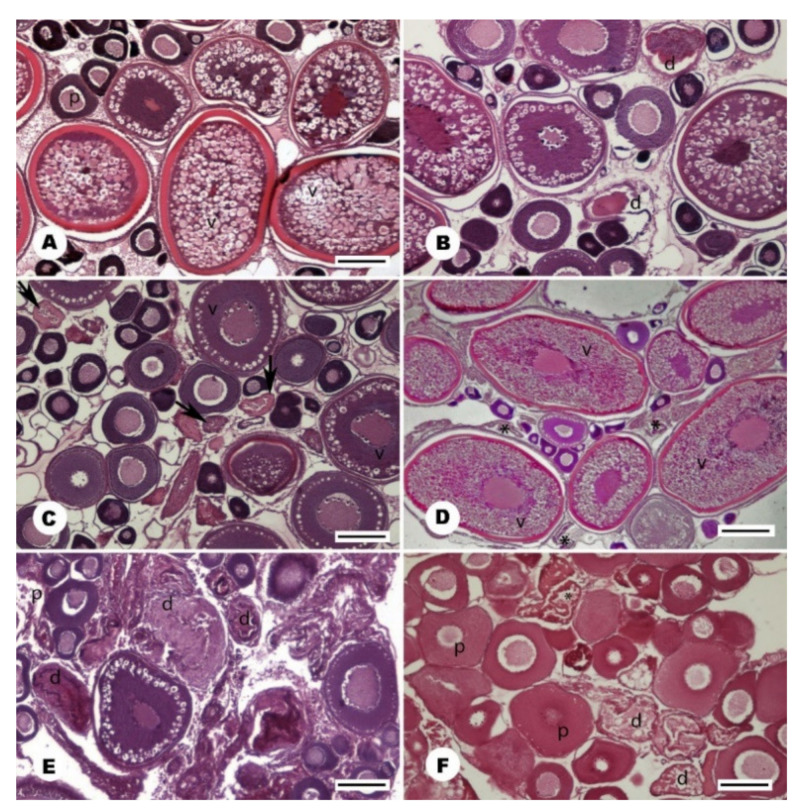
Histology of the gonad in the female stone moroko (*Pseudorasbora parva*) inhabiting Wardynka river, Poland. (**A**) a gonad with oocytes in the previtellogenesis (p) and vitellogenesis (v), February, bar 200 µm; (**B**) with numerous degenerated oocytes (d), November, bar 100 µm; (**C**) with oocytes at the beginning of vitellogenesis (v) and with degenerating oocytes at previtellogenesis (arrow), September, bar 150 µm; (**D**) spawning gonads with oocytes in an advanced vitellogenesis (v), post-ovulatory follicles (asterisk) and the oocytes at the beginning of the next generation of previtellogenesis, June, bar 200 µm; (**E**) after spawning with the oocytes at the early previtellogenesis (p) and degenerated oocytes (d), July, bar 100 µm; (**F**) with degenerating oocytes (d) in previtellogenesis (p) and post-ovulatory follicles (asterisk), end of July, bar 100 µm.

**Figure 5 animals-11-02627-f005:**
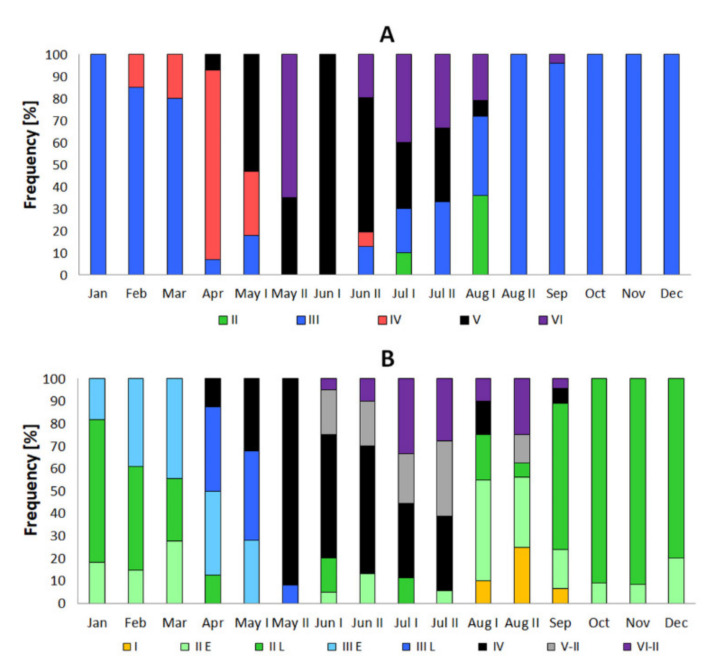
Frequency of gonad stage in stone moroko *Pseudorasbora parva* female (**A**) and male (**B**) inhabiting the Wardynka river, Poland.

**Figure 6 animals-11-02627-f006:**
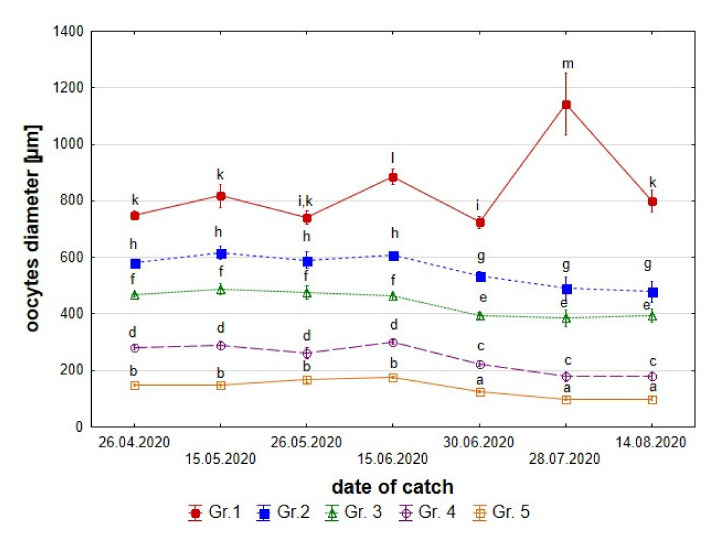
Average sizes of the five groups of stone moroko *Pseudorasbora parva* oocytes from April–August (see the text for the group description). Values with different letters show significant differences (*p* < 0.05; ANOVA test).

**Figure 7 animals-11-02627-f007:**
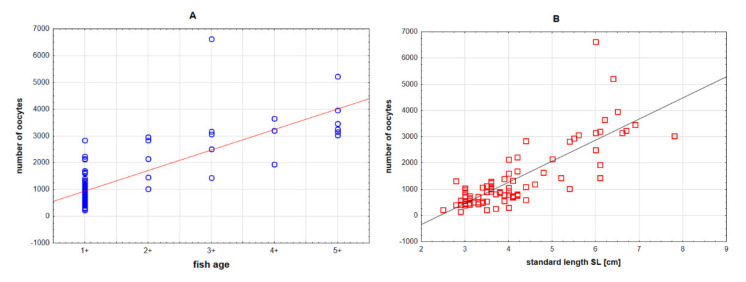
Relationship between number of oocytes and age (**A**) and standard length (**B**) in stone moroko *Pseudorasbora parva* inhabiting the Wardynka river, Poland.

**Figure 8 animals-11-02627-f008:**
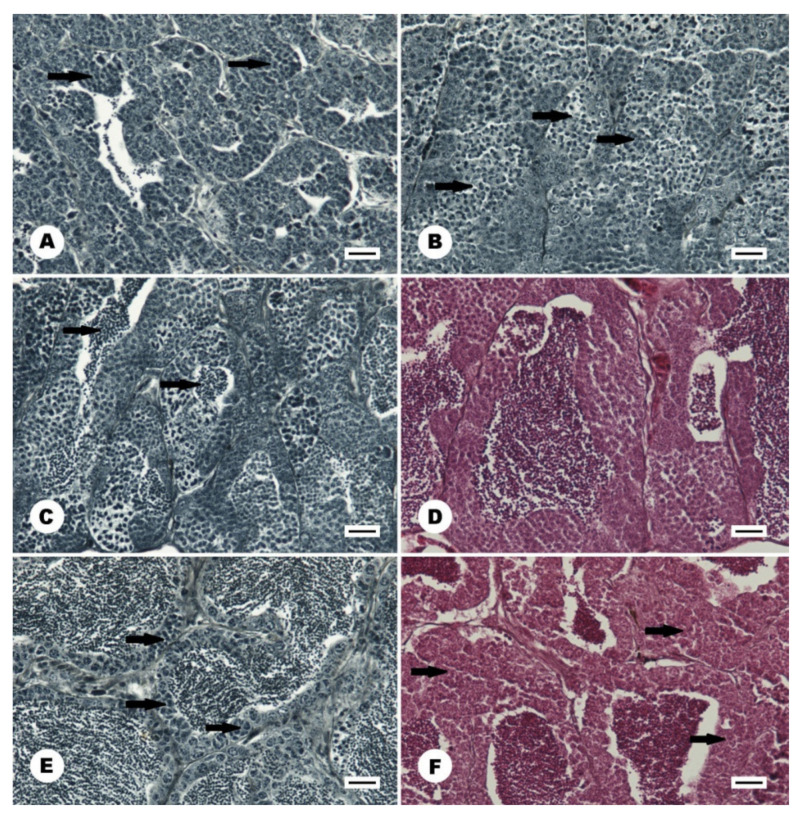
Histology of the gonad in the male stone moroko (*Pseudorasbora parva*) inhabiting Wardynka river, Poland. (**A**) gonad in later stage II_L_, the tubules contain numerous cyst with multiplying spermatogonia B (arrow), August; (**B**) Stage III_E_, the meiotic division has begun, and primary spermatocytes were formed in cyst (arrow), January; (**C**) Stage III_L_, all type of spermatogenic cells are visible in the gonad, first spermatozoa are formed (arrow), April; (**D**) The spawning stage IV, the tubule lumen is filled with spermatozoa, by the tubule wall numerous cysts with maturing cell occurred, April; (**E**) The spawning stage V-II, the gonad with finalised spermatogenesis, tubules filled with spermatozoa, the tubule wall contains resting type A spermatogonia and multiplying type B spermatogonia (arrow), June; (**F**) a spent gonad in stage VI-II. Shrunken tubules filled with unexpelled spermatozoa. The tubule wall contains type A spermatogonia and cysts with type B spermatogonia (arrow), June. Scale bar 20 μm.

**Table 1 animals-11-02627-t001:** The number of individuals of stone moroko, *Pseudorasbora parva*, used in the study.

Sex	Month												
	I	II	III	IV	V	VI	VII	VIII	IX	X	XI	XII	
	Number of Fish												Total
♀	8	13	10	14	49	49	13	20	24	13	12	9	234
♂	14	43	20	16	59	63	38	48	19	13	14	21	368

**Table 2 animals-11-02627-t002:** Morphological and biological characteristics of stone moroko, *Pseudorasbora parva* inhabiting the Wardynka river, Poland. Data are presented as mean ± SD and range.

Sex	Total Length (mm)	Standard Length (mm)	Fish Weight (g)	C_F_	GSI *	GSI **
♀Female n = 234	55.6 ± 14.6 30.0–93.0	46.2 ± 12.4 25.0–78.0	1.85 ± 1.66 0.20–8.29	1.44 ± 0.23 0.62–2.29	0.09–31.55	0.08–25.68
♂Male n = 368	63.4 ± 16.4 31.0–104.0	52.9 ± 14.1 28.0–87.0	3.30 ± 2.72 0.20–11.43	1.44 ± 0.24 0.25–3.24	0.07–6.98	0.06–6.93

* eviscerated fish, ** non-eviscerated fish.

**Table 3 animals-11-02627-t003:** Sizes of oocytes at different stages of vitellogenesis in stone moroko, *Pseudorasbora parva*. Data are presented as mean ± SD and range.

Size Oocytes (µm)				
Beginning of Previtellogenesis	Finally, Previtellogenesis	Beginning of Vacuolisation	First Occurrence of the Yolk	Completed Vitellogenesis
116.90 ± 17.75 ^a^ 67.50–145	229.70 ± 39.57 ^b^ 166.53–298.13	402.32 ± 51.57 ^c^ 315.21–495.12	668.47 ± 29.82 ^d^ 575.13–720.32	1026.87 ± 182.12 ^e^ 795.24–1725.01

SD = standard deviation; Values marked with different superscript letters show significant differences between the features (*p* < 0.05, ANOVA test).

**Table 4 animals-11-02627-t004:** Absolute and relative fecundity of the stone moroko, *Pseudorasbora parva*. Data are presented as mean ± SD and range.

Age					
1+	2+	3+	4+	5+	Total
absolute fecundity					
886 ± 506 ^a^ 225–2841	1538 ± 808 ^a^ 600–2952	3360 ± 1741 ^b^ 1443–6620	2945 ± 1341 ^a^ 1937–3649	3610 ± 363 ^c^ 3027–5220	1372 ± 1164 225–6620
relative fecundity					
1733 ± 778 ^a^ 268–4543	2241 ± 1307 ^a^ 1067–4119	1512 ± 518 ^a^ 1098–2390	1200 ± 503 ^a^ 697–1703	826 ± 118 ^a^ 707–988	1691 ± 832 268–4543

Values with different letters show significance differences (*p* < 0.05; ANOVA test).

## Data Availability

The data presented in this study are available on request from the corresponding author.

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
