# Peer review of "Reproductive Potential of Stone Moroko (Pseudorasbora parva, Temminck et Schlegel, 1846) (Teleostei: Cypriniformes: Gobionidae) Inhabiting Central Europe"

_animals, 2021, doi:10.3390/ani11092627_

Round 1

Reviewer 1 Report

The Authors of the revised manuscript entitled "Reproductive potential of stone moroko (Pseudorasbora parva, Temminck et Schlegel, 1846) (Teleostei: Cypriniformes: Gobionidae) an invasive fish inhibiting central Europe" have presented a very solid piece of study, based on excellent field work and meticulous histological analysis. The manuscript is a comprehensive analysis of a population of a highly freshwater species and presents interesting data, all of which should be beneficial especially for conservationists and ecologists. While the Introduction and Discussion were written very well and the presented data is quite abundant, there are several drawbacks and inaccuracies or omissions in the M&Ms and Results sections, which surely require to be corrected.

Before I get into details, in regard to reference formatting I would suggest to change it the way it is done in other MDPI papers: if there are more than 2 citations with sequential numeration, then they may be shortened, eg. in Line 69 instead of [15,16,17,24] it should be [15-17,24]. Please apply this principle throughout the manuscript.

Below I have outlined all my remarks, paragraph by paragraph.

Title: I believe it was supposed to be "inhabiting".

Simple Summary: Well written paragraph.
Line 23: Correct to "Europe".

Abstract: Again, well composed.
Line 37: Change to past tense "laid".

Introduction: This section is concise, presenting all the necessary facts for the average reader to understand the outlines of the study. Well done.
Line 48: Change "the taxon" to "this fish".
Line 69: Change "were" to "was".
Line 76: Change to past tense "subjected".

Materials and Methods: This part is mostly comprehensive, but there are some issues needing clarification. Furthermore, apart from the objections below, I would suggest to split the M&Ms into numbered subsections, such as: "Location", "Sampling and slide preparation", "Histology of female gonads", "Histology of male gonads", "Statistical analysis".
Line 85: Please specify in parenthesis which months included the two-week sampling interval (reproductive season).
Line 92: I am quite certain that the Authors calculated the Fulton's condition factor (CF), which is why there should be "body mass" instead of "gonad mass" in the parenthesis.
Line 94: The gonads of all of 602 sampled fish were subdued to histological analysis?
Figure 1: Please improve the resolution of this picture, the names of the river/lakes are barely readable. Is there a source of this map or is it the Authors' own composition?
Table 1: Please improve the formatting of (missing) lines and the proper alignment in rows of the three "Total" numbers.
Line 110: The use of the stereomicroscope is a bit unclear here - does it mean that microscope slides of gonads were studied under a stereomicroscope too?
Line 124: Change to plural "females".
Lines 134, 137 and 138: Change to adjective "follicular".
Line 138: To avoid confusion, I believe that "Stage 4 stadium" should be replaced with "class 4", as it is a reference to the previously described maturation groups/classes of oocytes, and not the currently discussed stages of the whole gonadal cycle.
Lines 140-172: The stages II and III of the male gonads have substages "E" (early) and "L" (late), while stages V and VI have substages "I" and "II". I wonder, why wasn't this nomenclature unified?
Line 175: Here, the Authors mention "investigated sites", but only one site was introduced at the start of M&Ms. Please explain this or correct if this was a typo.

Results: There are plenty of results in this paper, and while most of them were presented properly, there are some inconsistencies, some of which likely originate from missing methodology in the M&Ms. Specific commentary below.
Paragraph 3.1: There are water parameters indicated, but in M&Ms there was not a word about the methodology of these measurements. Please fix this in the M&Ms, likely in the first subsection.
Table 2: Why is the eviscerated GSI not shown here and the regular GSI only shown as ranges, not means with SD?
Line 194: Add "ranged" after "age".
Line 201: Correct to singular "stage".
Figure 3: The Authors indicated earlier that they sampled the fish twice a month during the breeding season, but this graph ignores it and shown only the monthly averages, while below in the text (Line 214) it is specifically indicated that "in second half of May ...". I understand that the visually this graph would become maybe too dense, but I believe this data could be still presented in some way.
Figure 4: Correct in Line 247 the letter for the last picture (in the description its "E" instead of "F").
Line 260: The Authors mention the use of ANOVA, but it has never been mentioned in the M&Ms sections. Where was it used and to which parameters? Please correct this.
Line 268: Start the sentence with "In".
Figure 6: There is something wrong with the letters describing the statistically significant differences - why are there so many letters (a-p) and not multiple letters (in some cases) as in Figure 3? Is "p" truly different than "e" for May? How about the two "L", which have totally different values but are not different? Please fix these issues.
Table 4: Why was there no statistical analysis of these fecundity data? Due to the high disparity in each group?
Line 278: Change to plural "numbers".
Figure 7: There was no indication of correlation calculations in the Statistical analysis section in the M&Ms, please add such information. Besides, I suggest to add "from size groups 1-4 of" in the description after "oocytes".
Line 288: I believe this is a female aged "5+", as it can be seen in Figure 7A. Therefore, the female mentioned later is not "of the same age". Furthermore, why is there no mention of the female with the largest number of size 1-4 oocytes (age 3+, SL 6 mm, >6500 oocytes)?
Line 296: Correct to past tense "reached".
Line 306: Change "constitute" to "were found in". Later on, the next sentence (ending with "Figure 8D") is likely missing some words or there are some other mistakes, but it is tough to discern what was meant here.
Line 309: Change to past tense "constituted" and "predominated".
Line 316: Change "male leaking" to "males were leaking".
Line 317: Change "has spent gonad" to "had spent gonads".
Line 318: "Change "single" to "few".
Figure 8: The description states that there are arrows in 5 of the subpictures, but none of them are visible. Please correct that. Delete "a" in line 333.

Discussion and Conclusions: Apart from some minor linguistic mistakes (I did not list them here, but it is mostly the improper usage of singular/plural forms here and there or missing commas) these parts were written well and I have no meritoric objections. Please remove the random and unnecessary bold/italic words.
Line 389: Delete the unnecessary parenthesis with the citation (Giurca ...), it is already cited as [20].
Line 424: Add "was observed" at the end of the sentence (infront of [26]).
Line 430: The start of the sentence has a duplication: delete "Male of stone moroko".
Line 441: Add "were observed" before "until".
Line 469: Start the sentence with "In".

Author Contributions: Delete the template part "For research articles with several authors, a short paragraph specifying their individual contributions must be provided. The following statements should be used"

Author Response

Dear reviewer

Thank you for the reviews and valuable tips.

I enclose the list of changes made in our manuscript.

The Authors of the revised manuscript entitled "Reproductive potential of stone moroko (Pseudorasbora parva, Temminck et Schlegel, 1846) (Teleostei: Cypriniformes: Gobionidae) an invasive fish inhibiting central Europe" have presented a very solid piece of study, based on excellent field work and meticulous histological analysis. The manuscript is a comprehensive analysis of a population of a highly freshwater species and presents interesting data, all of which should be beneficial especially for conservationists and ecologists. While the Introduction and Discussion were written very well and the presented data is quite abundant, there are several drawbacks and inaccuracies or omissions in the M&Ms and Results sections, which surely require to be corrected.- The Material and Metods and Results sections  were rewritten

Before I get into details, in regard to reference formatting I would suggest to change it the way it is done in other MDPI papers: if there are more than 2 citations with sequential numeration, then they may be shortened, eg. in Line 69 instead of [15,16,17,24] it should be [15-17,24]. Please apply this principle throughout the manuscript. -the range of numer was used to citing the article

Below I have outlined all my remarks, paragraph by paragraph.

Title: I believe it was supposed to be "inhabiting". -the letter was changed. The v-2 of manuscript was corrected by native speaker

Simple Summary: Well written paragraph.
Line 23: Correct to "Europe".-  the mistake was corrected

Abstract: Again, well composed.
Line 37: Change to past tense "laid". –the word was changed

Introduction: This section is concise, presenting all the necessary facts for the average reader to understand the outlines of the study. Well done.
Line 48: Change "the taxon" to "this fish".- the word was changed
Line 69: Change "were" to "was".- the fragment was changed accorging reviewer no 3 for „has been conducted”
Line 76: Change to past tense "subjected".- the word was changed

Materials and Methods: This part is mostly comprehensive, but there are some issues needing clarification. Furthermore, apart from the objections below, I would suggest to split the M&Ms into numbered subsections, such as: "Location", "Sampling and slide preparation", "Histology of female gonads", "Histology of male gonads", "Statistical analysis".- four subsections were used in the manuscript because of gonad maturity scale was deleted in accordance with the reviewer no 3 suggestion
Line 85: Please specify in parenthesis which months included the two-week sampling interval (reproductive season).- the period was inserted
Line 92: I am quite certain that the Authors calculated the Fulton's condition factor (CF), which is why there should be "body mass" instead of "gonad mass" in the parenthesis. - the mistake was corrected
Line 94: The gonads of all of 602 sampled fish were subdued to histological analysis? - yes
Figure 1: Please improve the resolution of this picture, the names of the river/lakes are barely readable. Is there a source of this map or is it the Authors' own composition? -the Figure 1 was improved
Table 1: Please improve the formatting of (missing) lines and the proper alignment in rows of the three "Total" numbers. -the table was improved
Line 110: The use of the stereomicroscope is a bit unclear here - does it mean that microscope slides of gonads were studied under a stereomicroscope too? -histological slide was analysed under microscope Nikon Eclipse 80i. The Material and Methods was rewritten
Line 124: Change to plural "females". -was chaned to plular number
Lines 134, 137 and 138: Change to adjective "follicular".- the paragraf was deleted
Line 138: To avoid confusion, I believe that "Stage 4 stadium" should be replaced with "class 4", as it is a reference to the previously described maturation groups/classes of oocytes, and not the currently discussed stages of the whole gonadal cycle. - The groups/classes concept consider oocyte development while stage concept consider gonad maturity
Lines 140-172: The stages II and III of the male gonads have substages "E" (early) and "L" (late), while stages V and VI have substages "I" and "II". I wonder, why wasn't this nomenclature unified? -Substage „late and early” (used  for of II and III stages) discribe the sequence of transformation of spermatogonia (stage II E or II L) or spermatocytes (stage stage I/iI E or III L)  while breaking of stage  V and VI  (V-I, V-II,stage VI-I and VI-II and VI-III) marking the overlapping of the stages of develoment. this is a consequence of not expeled large amount of spermatozoa. Early and late substage always occurs one by one while  overlapping of the stages doas not always occurs. The overlapping uses symbols of the stages of maturity development.
Line 175: Here, the Authors mention "investigated sites", but only one site was introduced at the start of M&Ms. Please explain this or correct if this was a typo.- a single numer was used

Results: There are plenty of results in this paper, and while most of them were presented properly, there are some inconsistencies, some of which likely originate from missing methodology in the M&Ms. Specific commentary below.
Paragraph 3.1: There are water parameters indicated, but in M&Ms there was not a word about the methodology of these measurements. Please fix this in the M&Ms, likely in the first subsection.- the method for physicochemical parameters of water measurement was inserted
Table 2: Why is the eviscerated GSI not shown here and the regular GSI only shown as ranges, not means with SD? – The Table 2 was modified: gonad weight was deleted and GSI for non-eviscerated and eviscerated fish was placed. For GSI only range was shown cause the value fluctuated across the year following maturity of gonad  (the mean and SD is an unreasonable value for calendar year)
Line 194: Add "ranged" after "age".- the word was added
Line 201: Correct to singular "stage". -singular numer was used
Figure 3: The Authors indicated earlier that they sampled the fish twice a month during the breeding season, but this graph ignores it and shown only the monthly averages, while below in the text (Line 214) it is specifically indicated that "in second half of May ...". I understand that the visually this graph would become maybe too dense, but I believe this data could be still presented in some way.  - two-weekly data in the spawning period were placed  the Figure 3 and 5
Figure 4: Correct in Line 247 the letter for the last picture (in the description its "E" instead of "F").  –the  letter was corrected
Line 260: The Authors mention the use of ANOVA, but it has never been mentioned in the M&Ms sections. Where was it used and to which parameters? Please correct this. – The statistical test was corrected in Material and Methods
Line 268: Start the sentence with "In" - „in” was inserted
Figure 6: There is something wrong with the letters describing the statistically significant differences - why are there so many letters (a-p) and not multiple letters (in some cases) as in Figure 3? Is "p" truly different than "e" for May? How about the two "L", which have totally different values but are not different? Please fix these issues. letter marks of statistical test was improved
Table 4: Why was there no statistical analysis of these fecundity data? Due to the high disparity in each group? - letter marks of statistical test was inserted
Line 278: Change to plural "numbers"- the plural numer was used
Figure 7: There was no indication of correlation calculations in the Statistical analysis section in the M&Ms, please add such information. Besides, I suggest to add "from size groups 1-4 of" in the description after "oocytes".- the calculation was added in Materials and Methods
Line 288: I believe this is a female aged "5+", as it can be seen in Figure 7A. Therefore, the female mentioned later is not "of the same age". Furthermore, why is there no mention of the female with the largest number of size 1-4 oocytes (age 3+, SL 6 mm, >6500 oocytes)?. Two sentences were deleted while female aged 3+ with the largest numer of oocytes was mentioned in the Result Section.
Line 296: Correct to past tense "reached". -tense  was corrected
Line 306: Change "constitute" to "were found in". Later on, the next sentence (ending with "Figure 8D") is likely missing some words or there are some other mistakes, but it is tough to discern what was meant here. -sentence was corrected
Line 309: Change to past tense "constituted" and "predominated". -words were corrected
Line 316: Change "male leaking" to "males were leaking".- sentence was corrected
Line 317: Change "has spent gonad" to "had spent gonads". -sentence was corrected
Line 318: "Change "single" to "few". -sentence was corrected
Figure 8: The description states that there are arrows in 5 of the subpictures, but none of them are visible. Please correct that. Delete "a" in line 333. - the letter was deleted

Discussion and Conclusions: Apart from some minor linguistic mistakes (I did not list them here, but it is mostly the improper usage of singular/plural forms here and there or missing commas) these parts were written well and I have no meritoric objections. Please remove the random and unnecessary bold/italic words.- the style of letters was unified. The version no 2 of the manuscript was corrected by native speaker
Line 389: Delete the unnecessary parenthesis with the citation (Giurca ...), it is already cited as [20]. - the autors name was dedeted
Line 424: Add "was observed" at the end of the sentence (infront of [26]).- the sentence was rewrtten
Line 430: The start of the sentence has a duplication: delete "Male of stone moroko". - the duplication was deleted
Line 441: Add "were observed" before "until". - the word was inserted
Line 469: Start the sentence with "In". - sentence was changed according the suggestion

Author Contributions: Delete the template part "For research articles with several authors, a short paragraph specifying their individual contributions must be provided. The following statements should be used" - the introductory part of the paragraph has been removed

Reviewer 2 Report

This manuscript is well written. I only have some editorial comments.

I suppose the fish species is from East Asia not South-East Asia.

L. 23: Europy must be Europe.

Caption of Figure 4: last photo (E) must be (F).

Upper case is used for oocyte phase but it confusing. So, lower case should be used.

Author Response

Dear reviewer

Thank you for the reviews and valuable tips.

I enclose the list of changes made in our manuscript.

I suppose the fish species is from East Asia not South-East Asia.- the region was specified only to the East Asia

L 23: Europy must be Europe.- the letter was changed. The version no 2 of the manuscript was corrected by native speaker

Caption of Figure 4: last photo (E) must be (F). - the letter was changed

Upper case is used for oocyte phase but it confusing. So, lower case should be used.- the upper case was repleced by lower case

Reviewer 3 Report

General comments

In their manuscript, Kirczuk and colleagues study an invasive population of stone moroko in central Europe and its reproductive potential. The manuscript must be reviewed by an English native speaker; writing is quite sloppy and full of errors. I hereby note some of them. Furthermore, the results are presented as unique observations rather than statistically sound outcomes. The manuscript needs substantial revision.

Specific comments

Title: Maybe the authors mean inhabiting.

Line 18: change originally to originated

Line 23: Europe

Line 25: change maturity to maturation

Line 44: Cypriniformes

Line 69: have been conducted

Line 74: selected

Line 76: subjected

Materials and methods:

Line 86: Please note MS-222 concentration and manufacturer

Lines 126-172: The description of stages can be omitted since they are already published and any minor modification should be referred.

Lines 174-181: Why non-parametric tests used for parametric data such as standard length? Did normality and equality of variance not met? Please clarify.

Lines 184-187: How physicochemical parameters were measured. Please clarify.

Results

Line 239: The calculation of 25mm as the smallest female attempting spawning is a random observation of the specific data. The authors should calculate the probability of the 50% percent mature, with a probabilistic model, a.k.a. size at the onset of maturity.

Lines 277-278: Rephrase the sentence

Lines 282-283: Rephrase

Line 413: part of the sentence is in bold

Figure1: Provide a picture with a better resolution

Author Response

Dear reviewer

Thank you for the reviews and valuable tips.

I enclose the list of changes made in our manuscript.

Specific comments

Title: Maybe the authors mean inhabiting.- the word was changed. The version no 2 of the manuscript was corrected by native speaker

Line 18: change originally to originated- the word was changed

Line 23: Europe – the word was changed

Line 25: change maturity to maturation– the word was changed

Line 44: Cypriniformes– the word was changed

Line 69: have been conducted-  the sentence was changed

Line 74: selected -the word was changed

Line 76: subjected -the word was changed

Materials and methods

Line 86: Please note MS-222 concentration and manufacturer - the specification have been inserted

Lines 126-172: The description of stages can be omitted since they are already published and any minor modification should be referred - the description of the maturity stage was deleted and the scale modification was pointed

Lines 174-181: Why non-parametric tests used for parametric data such as standard length? Did normality and equality of variance not met? Please clarify –  The statistical test was corrected for one-way ANOVA. Normality and  equality allowed for use parametric test.

Lines 184-187: How physicochemical parameters were measured. Please clarify. – the method was added in the M&Ms section

Results

Line 239: The calculation of 25mm as the smallest female attempting spawning is a random observation of the specific data. The authors should calculate the probability of the 50% percent mature, with a probabilistic model, a.k.a. size at the onset of maturity. Our work concerns the sexual cycle of mature fish (only numbers and data of mature fish are given in the manuscript). In the caught samples there were numerous immature specimens. Based on the backround the size of the smallest maturing fish and persent of mature individuals (consider sapwning period) were given in the manuscript

Lines  277-278: Rephrase the sentence – the sentence was rewritten

Lines 282-283: Rephrase– the sentence was rewritten

Line 413: part of the sentence is in bold – the style was unified

Figure1: Provide a picture with a better resolution – the Figure resolution was improved

Round 2

Reviewer 1 Report

The Authors of the revised manuscript entitled "Reproductive potential of stone moroko (Pseudorasbora parva, Temminck et Schlegel, 1846) (Teleostei: Cypriniformes: Gobionidae) an invasive fish inhabiting central Europe" have thoroughly followed instructions issued by all reviewers, as evidenced by the quality of the newly drafted paper.

All of the problems were fixed, all small mistakes corrected, all omissions added. Personally, I am very satisfied with the improvements which were done in regard to the statistical analyses and different figures. All of them are adequate now and mistakes have been eliminated.

Summing up, all of the introduced changes resulted in a comprehensively upgraded article, which now properly introduces the Reader to the high-quality work which has been done while conducting this research. Well done.

Author Response

Dear reviewer,

Thank you for your earlier comments that helped to improve our manuscript, as well as for kind words about the corrections made by us in our manuscript following the reviewers indications.

We did what we could with the English improvement  by native speaker. Minor adjustments were made by Guest Editor.

Sincerely yours

Katarzyna Dziewulska